# Characterization of Mn_5_Ge_3_ Contacts on a Shallow Ge/SiGe Heterostructure

**DOI:** 10.3390/nano14060539

**Published:** 2024-03-19

**Authors:** Troy A. Hutchins-Delgado, Sadhvikas J. Addamane, Ping Lu, Tzu-Ming Lu

**Affiliations:** 1Sandia National Laboratories, Albuquerque, NM 87185, USA; saddama@sandia.gov (S.J.A.); plu@sandia.gov (P.L.); tlu@sandia.gov (T.-M.L.); 2Center for Integrated Nanotechnologies, Albuquerque, NM 87185, USA; 3Center for High Technology Materials, University of New Mexico, Albuquerque, NM 87106, USA

**Keywords:** Mn5Ge3, Ge/SiGe, thin film, germanide, phase formation, solid-state synthesis

## Abstract

$Mn_{5}Ge_{3}$ is a ferromagnetic phase of the Mn-Ge system that is a potential contact material for efficient spin injection and detection. Here, we investigate the creation of $Mn_{5}Ge_{3}$-based contacts on a Ge/SiGe quantum well heterostructure via solid-state synthesis. X-ray diffraction spectra fitting indicates the formation of $Mn_{5}Ge_{3}$-based contacts on bulk Ge and Ge/SiGe. High-resolution scanning transmission electron microscopy imaging and energy dispersive X-ray spectroscopy verify the correct $Mn_{5}Ge_{3}$-based phase formation. Schottky diode measurements, transmission line measurements, and Hall measurements reveal that $Mn_{5}Ge_{3}$-based contacts serve as good p-type contacts for Ge/SiGe quantum well heterostructures due to having a low Schottky barrier height of 0.10
eV (extracted from a $Mn_{5}Ge_{3}$/n-Ge analogue) and a contact resistance in the order of 1 kΩ. Furthermore, we show that these electrical characteristics have a gate-voltage dependence, thereby providing tunability.

## 1. Introduction

Holes in germanium (Ge) have been shown to be an excellent candidate for spin-based alternative computing technologies, such as spin field-effect transistors (SpinFETs) for classical computing [1,2,3] and hole spin qubits for quantum computing [4,5,6,7,8,9,10]. In either technology, it is paramount to start with the best possible material. Recently, there have been demonstrations of Ge/SiGe quantum wells with mobilities in the $1\times 10^{6}$ c$m^{2}$/$V^{-1}$/$s^{-1}$ range [11,12]. Large-scale wafer growth of high-quality Ge/SiGe quantum materials has now opened access for further research [12,13].

Germanium has an advantage whereby Rashba spin-orbit coupling (SOC) is dominant over Dresselhaus SOC, thereby allowing for gate-tunable SOC. A gate-tunable SOC is crucial for SpinFET operation because it allows for direct control over the spin precession rate. However, control over spin precession is challenging if the transport is diffusive within the spin-orbit length. Fortunately, shallow and undoped germanium quantum wells have been shown to meet these requirements [14]. For SpinFETs, it is also important to have good spin injection and detection efficiencies. Optical spin injection in Ge/SiGe quantum wells has been theoretically shown to have 96% spin polarization [15] and experimentally shown to have 85% spin polarization [16]. While optical spin injection has high degrees of spin polarization, it is simply not trivial to extend to integrated systems for spintronic applications as compared with electrical spin injection.

Electrical spin injection is usually performed with a ferromagnetic contact that has either a conductivity mismatch with the substrate [17,18] or a tunnel junction [19,20,21,22,23]. $Mn_{5}Ge_{3}$ has been theoretically [24] and experimentally shown to be a good ferromagnetic contact for spin injection and detection in both p-type [1,2] and n-type [25] Ge. The easiest method to form $Mn_{5}Ge_{3}$ contacts is through solid-state synthesis whereby Mn thin films are deposited on Ge and annealed to form $Mn_{5}Ge_{3}$. However, to the best of our knowledge, $Mn_{5}Ge_{3}$ solid-state synthesis has not been directly applied to strained Ge/SiGe quantum well heterostructures. Here, we study the formation of $Mn_{5}Ge_{3}$-based contacts on Ge/SiGe through structural characterization and evaluate their electrical performance. Our study reveals the complexity of a solid-state reaction that can occur with forming $Mn_{5}Ge_{3}$ directly on a Ge/SiGe heterostructure. Our unique results thus provide a foundation for future studies regarding the solid-state synthesis of $Mn_{5}Ge_{3}$ contacts for spin injection/detection on high-quality Ge/SiGe material.

## 2. Materials and Methods

Mn thin films were deposited onto 1 c$m^{2}$ die of bulk (100) Ge and Ge/SiGe heterostructures via thermal evaporation of Mn powder immediately after surface cleaning. The surface cleaning procedure consists of standard solvent clean, oxygen plasma clean, and three cycles of buffered oxide etchant (BOE) and deionized water (DI $H_{2}O$) rinse [26]. The Ge/SiGe heterostructures used consist of a 15 nm Ge quantum well at a depth of 30 nm with $Si_{0.15}Ge_{0.85}$ barriers and a Si capping layer with an approximate thickness of 1 nm. Details about the growth and transport properties of the Ge/SiGe heterostructure have been previously reported [13]. The deposition of approximately 100 nm of Mn occurred at a pressure of $1\times 10^{-6}$ torr and a rate of 0.5 Å/$s^{-1}$. The die was then annealed at various temperatures for 30 min in a flash lamp annealer with Ar atmosphere. A Mn thickness of 100 nm was selected to ensure that there was enough Mn for the solid-state reaction to reach past the quantum well.

## 3. Results and Discussion

### 3.1. X-ray Diffraction Analysis

Mn thin films on bulk Ge were investigated first to verify that our process could result in $Mn_{5}Ge_{3}$ phase forming before proceeding to the Ge/SiGe heterostructures. Studies have shown that $Mn_{5}Ge_{3}$ formed via solid-state synthesis on (100) Ge and can occur from 250 °C up to 450 °C [27,28]. Therefore, we chose to proceed with 50 °C increments from 200 °C to 400 °C to deduce the appropriate annealing temperature for our annealing setup. X-ray diffraction (XRD) analysis of θ−2θ scans was performed to investigate the various phases of the Mn-Ge system formed at each of the annealing temperatures. The resulting θ−2θ scans and their analyses are shown in Figure 1. A detailed summary of the XRD analysis is represented in Table 1. Table 1 highlights the various phases found with corresponding 2θ, planes, lattice parameters (d-values), unit cell volume, and crystallite sizes extracted from the prominent peak of each phase at each annealing temperature. Crystallite sizes were determined using the Scherrer equation: (1)$$\tau =0.94\frac{\lambda }{\beta cos(\theta )}.$$

Here, τ is the crystallite size in Å, λ is the X-ray wavelength in Å, β is the full width at half max in radians, and θ is the diffraction angle. The XRD system we use has a wavelength of 1.54059 Å.

The XRD results for Mn on bulk Ge are summarized in Figure 1a, and fitting was performed with the International Centre for Diffraction Data (ICDD) database, PDF-2 [29]. When annealed at 200 °C (black curve), the film was determined to be a Mn-α phase. The Mn-α phase’s characteristic feature is the peak at 43° corresponding to the (411) plane. Annealing at 200 °C resulted in a Mn-α crystallite size of 171.08 Å. Similar to annealing at 200 °C, annealing at 250 °C (red curve) and 300 °C (blue curve) also resulted in the Mn-α phase with crystallite sizes of 186.24 Å and 221.95 Å, respectively. However, at 300 °C, the peak around 43° started to diminish, and smaller signatures started to appear at both higher and lower diffraction angles. These observed deviations indicate that Mn started to react with Ge to form $Mn_{5}Ge_{3}$ as this is the first phase known to form [27,28]. Next, the sample annealed at 350 °C (green curve) showed the same signatures that started to appear in the 300 °C–anneal sample with higher intensities. Furthermore, the characteristic peak of the Mn-α phase at 43° disappeared and was replaced by two peaks at around 42.35° and 43.73° corresponding to the $Mn_{5}Ge_{3}$ (211) and (112) planes, respectively. Additionally, there were peaks at 35.46°, 38.24°, 38.38°, and 46.22° that were only prominent at 350 °C, which arose from the (002), (210), (102), and (202) $Mn_{5}Ge_{3}$ planes, respectively. The peaks of the 350 °C sample fit to $Mn_{5}Ge_{3}$, confirming that $Mn_{5}Ge_{3}$ indeed started to form at 300 °C. The prominent peak for $Mn_{5}Ge_{3}$ was at 42.35°, by which crystallite sizes of 376.85 Å, 370.46 Å, and 512.94 Å were extracted, respectively, at 300 °C, 350 °C, and 400 °C. By 400 °C (magenta curve), most $Mn_{5}Ge_{3}$ peaks disappeared, and the new peaks fit to $Mn_{1}1Ge_{8}$. A crystallite size of 333.90 Å was obtained from the $Mn_{1}1Ge_{8}$ prominent peak at 42.73°. Based on the XRD results above, these annealing conditions can be transferred to Ge/SiGe heterostructures to form $Mn_{5}Ge_{3}$ and likely within the narrower range of 300–400 °C with the most promising temperature being 350 °C.

Figure 1b shows the XRD results for the annealed Mn films on the Ge/SiGe heterostructure and a reference scan (black curve) of Ge/SiGe. The Ge/SiGe heterostructures were annealed at 300 °C (red curve) and 400 °C (blue curve). Both the 300 °C and 400 °C data are similar with only the relative heights of peaks changing between the two. The XRD peaks best fit to ${\mathrm{Mn}}_{5}$${\left(Si_{0.3}Ge_{0.7}\right)}_{3}$, $Mn_{2}GeO_{4}$, and MnO. However, the ${\mathrm{Mn}}_{5}$${\left(Si_{0.3}Ge_{0.7}\right)}_{3}$ peaks do not perfectly align with the data. We expect phases to have the form ${\mathrm{Mn}}_{y}$ (${\mathrm{Si}}_{0.15}$${\mathrm{Ge}}_{0.85}$${)}_{x}$ as our material has ${\mathrm{Si}}_{0.15}$${\mathrm{Ge}}_{0.85}$ barriers, and this could explain the deviation from the fit. Nevertheless, XRD has detected the formation of the correct crystal structure type ${\mathrm{Mn}}_{5}$ (${\mathrm{Si}}_{x}$${\mathrm{Ge}}_{1-x}$${)}_{3}$, but microscopic verification is needed to confirm specifics about relative concentrations.

### 3.2. Scanning Transmission Electron Microscopy Analysis

We chose 350 °C as the optimal annealing temperature for making ${\mathrm{Mn}}_{y}$ (${\mathrm{Si}}_{0.15}$${\mathrm{Ge}}_{0.85}$${)}_{x}$ contacts since it was confirmed to be the best temperature for $Mn_{5}Ge_{3}$ on bulk Ge and gave a ±50 °C tolerance to the Ge/SiGe XRD results. We proceeded to make transmission line measurement (TLM) and Hall bar devices with ${\mathrm{Mn}}_{y}$ (${\mathrm{Si}}_{0.15}$${\mathrm{Ge}}_{0.85}$${)}_{x}$ contacts. An optical microscopy image in Figure 2a shows an example of the TLM (left) and Hall bar (right) devices after fabrication. The sample was used in electrical characterizations and microanalysis. Here, we discus the microanalysis results. High-resolution scanning transmission electron microscopy (HR-STEM) imaging, in combination with energy dispersive X-ray spectroscopy (EDS), was used to investigate the ${\mathrm{Mn}}_{y}$ (${\mathrm{Si}}_{0.15}$${\mathrm{Ge}}_{0.85}$${)}_{x}$ contacts in further detail. EDS provides a means to map elements in a structure by their respective Kα, Lα, and other X-ray signatures. The elements’ signatures are then represented by a false coloring in the STEM images. The STEM high-angle annular dark-field (HAADF) image in Figure 2b reveals a complex structure stack with not-well-defined boundaries. The regions labeled I, II, and III correspond to the contact structure that is of interest. The material below the region of interest is the ${\mathrm{Si}}_{0.15}$${\mathrm{Ge}}_{0.85}$ virtual substrate. The material atop the regions of interest correspond to the Au/Ti metal contact pads and the $Al_{2}O_{3}$ gate oxide. The EDS data in Figure 2c verify the clear distinction between the metal/oxide region and the complex contact regions. Here, Au is represented with green, blue corresponds to Ti, and red corresponds to Al (from $Al_{2}O_{3}$). Black represents the absence of any detected signatures. Figure 2d shows an overlay of Mn, O, and Ge maps, revealing the complexity of the structure. Here, green corresponds to Mn, blue is Ge, and red is O.

Focusing on region I, the Mn ( Figure 2e) and O ( Figure 2f) elemental maps show that region I is entirely composed of Mn and O and is approximately 70 nm thick. This claim is supported by the absence of Ge and Si in region I, as shown in their respective elemental maps in Figure 2g,h. The Mn and O maps provide supporting evidence that the broad peak in the XRD data indeed corresponds to oxidized Mn. Furthermore, region I was determined to be polycrystalline MnO through HR-STEM HAADF imaging. Region II, like region I, contains Mn and O. However, region II also contains Ge and is approximately 116 nm thick. Unlike any of the other regions, region II does not have a uniform distribution of elements. Together, the Ge (Figure 2g) and Si (Figure 2h) data highlight the complexity of region II. The Si map shows only trace amounts where there are large concentrations of Ge. The Ge map shows that there are islands of Ge mostly near the region I/region II interface. Interestingly, the Mn and O distributions are nearly uniform in region II except at the areas of highest Ge concentrations and where the EDS data do not correspond to any of the elements tested (black region). This black region, most easily visualized in Figure 2d, could indicate the presence of a void since there are none of the expected elemental signatures and a lack of resolvable features in the HAADF STEM image. The Mn and Ge maps indicate a low concentration likely coming from behind the void. While voids could still occur due to the diffusion of Ge as discussed, the formation here is likely due to the complexity of the reaction that occurred. A better-quality film will form when annealed in a high-vacuum environment, and voids will be less likely to form [30,31,32,33]. Besides the islands of Ge, region II is likely a mixture of MnO and ${\mathrm{Mn}}_{5}$${\mathrm{Ge}}_{3}$$O_{x}$ due to its location between region I and region III, which is consistent with previous reports of solid-state syntheses of $Mn_{5}Ge_{3}$ [33,34,35,36].

Region III, the largest of the three regions with a thickness of approximately 186 nm, appears to have the best film quality (as determined by HR-STEM) and a promising elemental map for ${\mathrm{Mn}}_{y}$ (${\mathrm{Si}}_{0.15}$${\mathrm{Ge}}_{0.85}$${)}_{x}$ contacts. The film has relatively uniform distributions of Mn, Ge, and dilute Si, as emphasized by the integrated EDS line profile in Figure 2i. Furthermore, the Si line is magnified such that when Si and Ge counts are equal, it represents the relative concentration of the barrier ${\mathrm{Si}}_{0.15}$${\mathrm{Ge}}_{0.85}$. The region II/III interface is where the Ge/SiGe heterostructure starts. While it is clear that Mn is being driven into the heterostructure, the presence of Ge in region II indicates that Ge has left the heterostructure, thus making the heterostructure more Si rich.

To further investigate the film, HR-STEM HAADF imaging along with its fast Fourier transform (FFT) pattern was used to determine the exact phase of ${\mathrm{Mn}}_{y}$ (${\mathrm{Si}}_{0.15}$${\mathrm{Ge}}_{0.85}$${)}_{x}$. The STEM HAADF image of ${\mathrm{Mn}}_{y}$ (${\mathrm{Si}}_{0.15}$${\mathrm{Ge}}_{0.85}$${)}_{x}$ in Figure 3a, along with its FFT pattern in Figure 3b, is in excellent agreement with the projected structure (Figure 3c) and simulated selected area electron diffraction (SAED) pattern (Figure 3d) corresponding to the hexagonal crystal structure of $Mn_{5}Ge_{3}$. The SAED pattern is calculated using reciprocal lattice and associated electron structure factors under the kinematical approximation [37]. The geometry of the pattern (or relative positions of the reciprocal lattice) is important in this context, and fits exactly the FFT pattern in [1, −2, 0], confirming the phase. Since the starting material consisted of mostly ${\mathrm{Si}}_{0.15}$${\mathrm{Ge}}_{0.85}$ layers, except for the Ge quantum well and Si cap, it is likely that the relative concentrations of Si and Ge remained near their original concentrations such that the film is ${\mathrm{Mn}}_{5}$ (${\mathrm{Si}}_{0.15}$${\mathrm{Ge}}_{0.85}$${)}_{3}$. The EDS line scan, calibrated to the ${\mathrm{Si}}_{0.15}$${\mathrm{Ge}}_{0.85}$ virtual substrate beyond region III, is consistent with this claim. The XRD data showed a fit with ${\mathrm{Mn}}_{5}$${\mathrm{Si}}_{0.9}$${\mathrm{Ge}}_{2.1}$ or ${\mathrm{Mn}}_{5}$ (${\mathrm{Si}}_{0.3}$${\mathrm{Ge}}_{0.7}$${)}_{3}$, and this could be due to the larger concentration of Si at the interface of region II and region III. Nevertheless, our data revealed the correct phase of ${\mathrm{Mn}}_{5}$ (${\mathrm{Si}}_{x}$${\mathrm{Ge}}_{1-x}$${)}_{3}$ to make a p-type ferromagnetic contact for Ge/SiGe heterostructure field-effect transistors (HFETs).

### 3.3. Electrical Characterization

With the confirmation of solid-state-reaction-formed $Mn_{5}Ge_{3}$-based contacts on undoped Ge/SiGe heterostructures, we present the electrical properties of the $Mn_{5}Ge_{3}$ contacts from Schottky diodes, TLM, and Hall bar measurements. Since it can be difficult to analyze Schottky contacts on quantum well heterostructures, our Schottky diodes were fabricated on n-type (100) Ge, with a resistivity range of 1–10 Ωcm, to estimate the Schottky barrier height of $Mn_{5}Ge_{3}$/Ge. $Mn_{5}Ge_{3}$/n-Ge Schottky diodes serve as a good analogue for estimating the barrier height for $Mn_{5}Ge_{3}$/Ge in our Ge/SiGe heterostructure devices. Ideally, there would be an increase in barrier height due to the compressive strain of the Ge quantum well moving both the conduction and valence band edges farther apart [38,39]. However, Ge is widely known to pin the Fermi levels of most metals very close to the valence band edge, thus nullifying any effects due to strain in the case of hole transport. The lack of strain affecting barrier heights due to Fermi level pinning has been shown in the case of n-type SiGe [40]. While $Mn_{5}Ge_{3}$/n-Ge is not a direct replacement for the $Mn_{5}Ge_{3}$/stained-Ge quantum well, it does provide the best estimate available. A direct measurement for our system would require contact only of the quantum well with $Mn_{5}Ge_{3}$ in an isolated fashion, which is extremely difficult, especially given its relatively small thickness. To fabricate the $Mn_{5}Ge_{3}$/Ge Schottky diodes, Mn films on Ge were etched back into circular contacts and then annealed to form $Mn_{5}Ge_{3}$. The Schottky diodes were made with a radius of 200 μm. The $Mn_{5}Ge_{3}$/Ge Schottky diodes were characterized using current density–voltage analysis at room temperature. The current density–voltage data are represented in Figure 4a. The Schottky barrier height is extracted from the following relation: (2)$$\frac{k_{B}T}{e}\ln\left(\frac{J}{A^{*}T^{2}}\right)\approx \frac{V}{n}-\frac{\phi }{e}.$$

Here, $\frac{k_{B}T}{e}$ is the voltage corresponding to the thermal energy in eV, $A^{*}$ is the Richardson constant, *T* is the temperature, *J* is the current density, *V* is the voltage, *n* is a nonideality factor, and $\frac{\phi }{e}$ is the Schottky barrier height. The red line in Figure 4d is the fit line where a Schottky barrier height, with respect to the Ge conduction band, of 0.57(1) eV is obtained. The fit assumes $A^{*}=$ 44.5(105) A/c$m^{-2}$/$K^{-2}$ and T= 300 K. Therefore, $Mn_{5}Ge_{3}$ has a Schottky barrier height of 0.10(1) eV with respect to the Ge valence band. The assumptions made and the obtained Schottky barrier height are consistent with other studies [1,2,25,32,41].

While the Schottky diode on bulk n-type Ge gives an estimate of the $Mn_{5}Ge_{3}$/Ge barrier height for the heterostructure, it does not provide a full scope of the contact’s electrical characteristics. Devices made with Ge/SiGe will be enhancement-mode heterostructure field-effect transistors (HFETs), whose characteristics will be dependent on applied gate voltages. Furthermore, the material was grown such that there was high-quality transport at cryogenic temperatures. Therefore, we fabricated TLM and Hall bar devices (as previously described [13], except with surface preparation, as described in this manuscript) to measure the $Mn_{5}Ge_{3}$ contacts’ properties under these conditions. TLM and Hall measurements were performed using four-terminal, low-frequency lock-in techniques at a temperature of 4 K using a liquid helium cryostat. A 100 μV sinusoidal excitation was sourced from a lock-in for both the TLM and Hall measurements. For Hall measurements, the magnetic field was swept from −0.5 to 0.5 T.

TLM devices provide a means to extract the contact and gate-induced 2D hole gas (2DHG) sheet resistances of materials. Our TLM devices (left device in Figure 2a) were made with 25 μm × 50 μm
$Mn_{5}Ge_{3}$–based contacts with separations (channel lengths) of 50 μm, 100 μm, 325 μm, 400 μm, and 675 μm. A 50 μm–wide gate that overlaps with all the contacts provides a means to capacitively induce 2DHG. A more negative gate voltage increases the 2DHG density and decreases the 2DHG sheet resistance. Contact resistance was obtained from the TLM device using two different methods, each using the TLM equation, assuming that the contact resistance is much larger than the resistance of the metal pads: (3)$$R_{T}=R_{S}\frac{L}{w}+2R_{C}.$$

Here, $R_{T}$ is the total measured resistance, $R_{S}$ is the 2DHG sheet resistance, $\frac{L}{w}$ is the ratio of the channel length to the channel width, and $R_{C}$ is the contact resistance. The first method used is the standard approach, where the total resistance is measured as a function channel length, a linear fit is performed, and the contact resistance is half the y-intercept of the linear fit. The 2DHG sheet resistance is then extracted by multiplying the fit’s slope by the channel width. The extracted contact resistance and 2DHG sheet resistance, as a function of gate voltage, are shown in Figure 4b,c, respectively. A contact resistance in the order of 1 kΩ across the measured voltage range was obtained. Interestingly, the 2DHG sheet resistance saturates at gate voltages more negative than −15 V, while the contact resistance has a linear dependence on gate voltage after the saturation of the device. The gate voltage dependence past saturation can be explained by the gate’s overlap with the $Mn_{5}Ge_{3}$-based contact. Although the saturation of holes has been achieved in the channel of the device, the electric field from the gate lowers the injection barrier. Thus, as the gate voltage is further increased past saturation, there is still a measured reduction in the contact resistance [42,43,44], while the 2DHG sheet resistance is no longer sensitive to the gate voltage. If there was no Schottky barrier at the $Mn_{5}Ge_{3}$/Ge interface in the Ge/SiGe HFET device, then both the 2DHG sheet resistance and the contact resistance would saturate, indicating an ohmic contact. An ohmic contact has little to no barrier such that charge transport is unaffected by any additional band bending at the metal/semiconductor interface. However, a Schottky barrier would be affected by further band bending. Either image-force lowering of the barrier height occurs or the barrier thickness is reduced. In either case, the result is an increase in the supply of carriers from an effectively reduced contact resistance [45].

To verify the behavior of the contact resistance past saturation, contact resistance was determined using a second method. The second method used involves subtracting off the sheet resistance component from the total resistance for each resistance measurement and dividing by 2 (for the two current contacts). This was made possible because the sheet resistance was obtained through a four-terminal measurement. The contact resistance obtained by the second method was then averaged across three different devices. The second method data (black) are shown in Figure 5a, where the error bars correspond to the standard deviation. The contact resistance obtained by the first method is overlaid in red. As expected, the two methods for extracting contact resistance were consistent with one another. The Hall density versus gate voltage data in Figure 5b show that a saturation density of 8 × 10^11^ cm^−2^ is obtained at a gate voltage of −15 V. The mobility versus density data in Figure 5c show that a peak mobility of 2.1 × 10^5^ cm^2^V^−1^s^−1^ is achieved at a density of 8 × 10^11^ cm^−2^. The transport data using $Mn_{5}Ge_{3}$-based contact are consistent with what was previously observed in the same Ge/SiGe heterostructures with PtSiGe-based contacts [13]. The saturation of Hall density with a higher gate voltage is characteristic of Ge/SiGe quantum well systems and is consistent with the saturation of the 2DHG sheet resistance. This further supports that the reduction in contact resistance is due to the capacitive coupling of the gate to the Schottky barrier.

## 4. Conclusions

We investigated $Mn_{5}Ge_{3}$-based contacts created via solid-state synthesis on a shallow Ge/SiGe quantum well heterostructure. XRD was used to determine the proper annealing temperature for the phase formation, while STEM HAADF and its FFT image were used to verify the formation of the target Mn-Ge phase. Although our solid-state synthesis could be improved to eliminate the unwanted oxides ($Mn_{2}GeO_{4}$ and MnO) observed, the $Mn_{5}Ge_{3}$-based contacts were observed to work well as p-type contacts for Ge/SiGe HFETs even at cryogenic temperatures. Our electrical characterization results were consistent with what is expected for $Mn_{5}Ge_{3}$ contacts and other p-type contacts on Ge/SiGe.

Our demonstration of the solid-state synthesis of $Mn_{5}Ge_{3}$-based contacts for a shallow Ge/SiGe quantum well heterostructure provides a clear path towards demonstrating spin transport in Ge 2DHG systems similar to what was done in Si 2DEG systems [46] and Ge nanowires [1,2]. The evidence of Schottky barrier lowering, shown from a decreasing contact resistance past saturation, is promising for efficient spin injection and its gate tunability [47]. Future studies should include studying the magnetic properties of ${\mathrm{Mn}}_{5}$(${\mathrm{Si}}_{0.15}$${\mathrm{Ge}}_{0.85}$${)}_{3}$ (formed after the removal of the top barrier and quantum well) and spin transport in the Ge/SiGe heterostructure.

## Figures and Tables

**Figure 1 nanomaterials-14-00539-f001:**
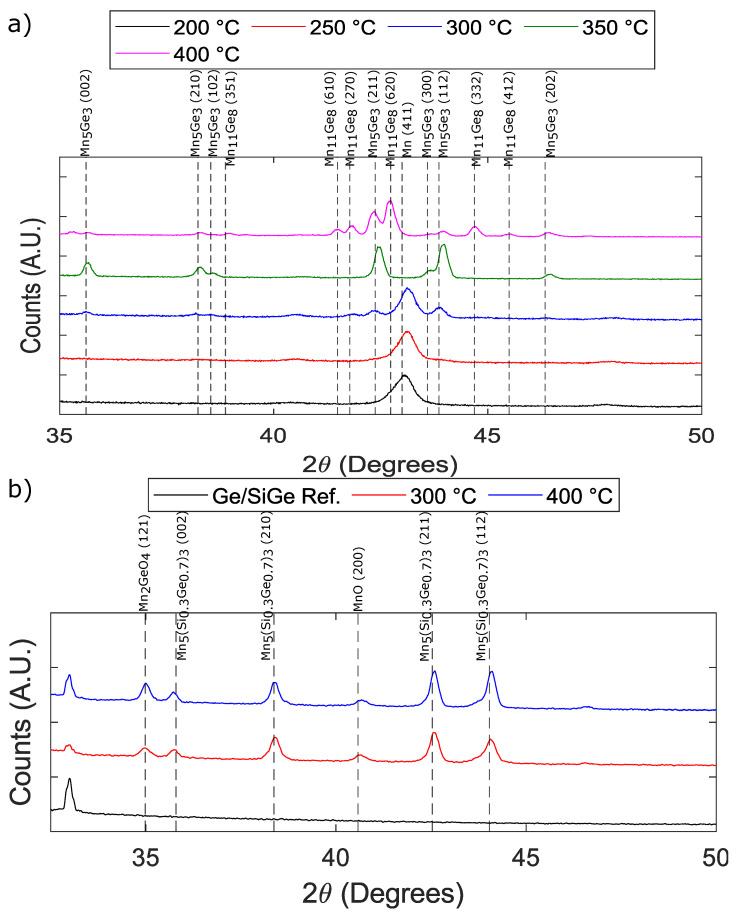
Summary of XRD θ−2θ scans. (**a**) XRD θ−2θ scans for thermal evaporated Mn thin films on (100) Ge substrates and annealed at 200 °C (black), 250 °C (red), 300 °C (blue), 350 °C (green), and 400 °C (magenta). The black vertical lines correspond to ICDD peak locations for $Mn_{5}Ge_{3}$ and $Mn_{1}1Ge_{8}$. (**b**) XRD θ−2θ scans for thermal evaporated Mn thin films on ${\mathrm{Ge}/\mathrm{Si}}_{0.15}$${\mathrm{Ge}}_{0.85}$ quantum well heterostructures and annealed at 300 °C (red) and 400 °C (blue). The ${\mathrm{Ge}/\mathrm{Si}}_{0.15}$${\mathrm{Ge}}_{0.85}$ reference scan is shown in black. The vertical lines correspond to ICDD peak locations for ${\mathrm{Mn}}_{5}$ (${\mathrm{Si}}_{0.3}$${\mathrm{Ge}}_{0.7}$${)}_{3}$ and various oxides: $Mn_{2}GeO_{4}$ and MnO.

**Figure 2 nanomaterials-14-00539-f002:**
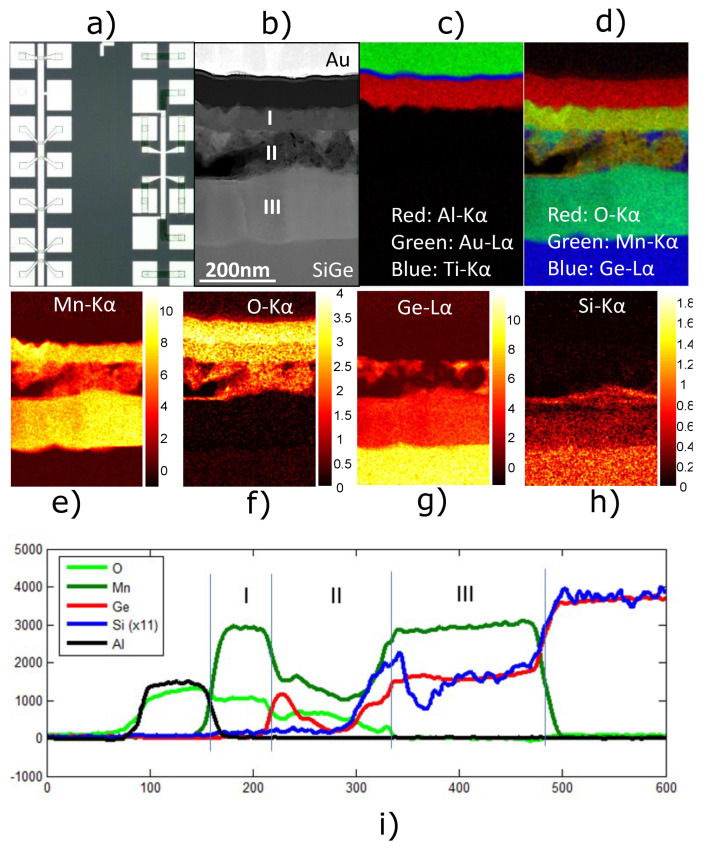
Cross-sectional STEM and EDS analysis of $Mn_{5}Ge_{3}$-based contacts for Ge/SiGe H-FET devices. (**a**) Optical microscope image TLM (**left**) and Hall bar (**right**) devices. (**b**) HAADF STEM image of the $Mn_{5}Ge_{3}$-based contact material stack. (**c**) Combined EDS map of the Au (red)/Ti (blue) gate metal and $Al_{2}O_{3}$ (Al-red) gate dielectric above the $Mn_{5}Ge_{3}$-based contact. (**d**) Combined EDS mapping of O (red), Mn (green), and Ge (blue). (**e**) EDS map of Mn, (**f**) EDS map of O, (**g**) EDS map of Ge, (**h**) EDS mapping of Si, and (**i**) integrated EDS line profile with signatures of O (light green), Mn (dark green), Ge (red), Si (blue), and Al (black).

**Figure 3 nanomaterials-14-00539-f003:**
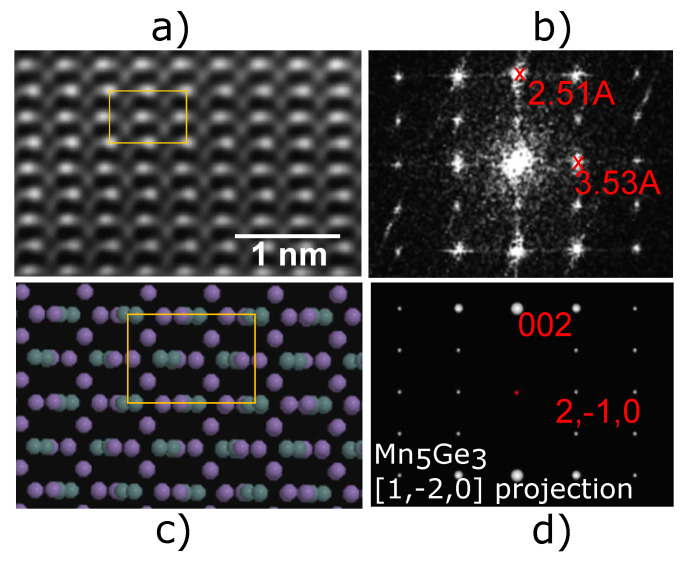
STEM HAADF analysis of $Mn_{5}Ge_{3}$-based contact. (**a**) HR-STEM HAADF of region III. (**b**) FFT of the STEM HAADF image. (**c**) Crystal structure of $Mn_{5}Ge_{3}$. (**d**) Simulated SAED pattern of $Mn_{5}Ge_{3}$ in the [1, −2, 0] projection.

**Figure 4 nanomaterials-14-00539-f004:**
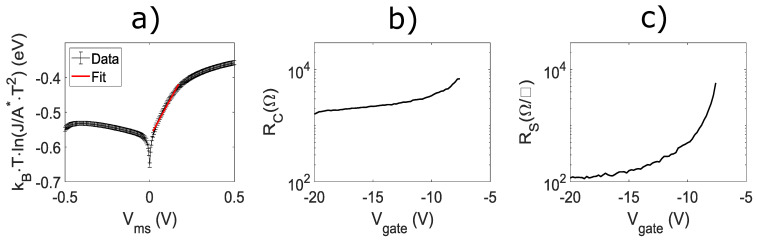
Electrical characterization of $Mn_{5}Ge_{3}$ contacts. (**a**) $Mn_{5}Ge_{3}$/n-type (100) Ge Schottky diode. (**b**) Contact resistance for TLM H-FET device. (**c**) Sheet resistance for TLM H-FET device.

**Figure 5 nanomaterials-14-00539-f005:**
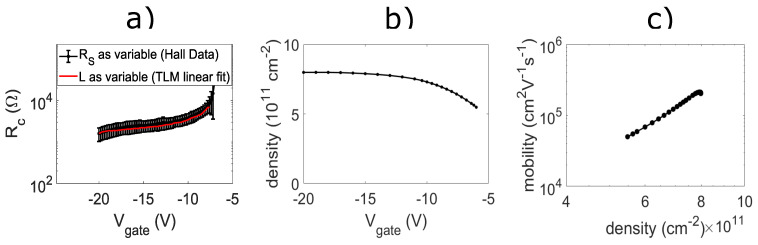
Transport characterization with $Mn_{5}Ge_{3}$ contacts. (**a**) Comparison of contact resistance obtained via Hall data (black) and TLM (red). (**b**) Hall density vs. gate voltage. (**c**) Mobility vs. density.

**Table 1 nanomaterials-14-00539-t001:** Table summary of X-ray diffraction analysis.

Phase	2θ	Plane	d-Value	Unit Cell Volume	Crystallite Size	Crystallite Size	Crystallite Size	Crystallite Size	Crystallite Size
		(h k l)	(Å)	(Å^3^)	(200 °C)	(250 °C)	(300 °C)	(350 °C)	(400 °C)
Mn-α	43.02°	4 1 1	2.201	707.85	171.08 Å	186.24 Å	221.95 Å	-	-
MnO	40.55°	2 0 0	2.223	87.88	-	-	271.51 Å	-	264.48 Å
$Mn_{2}GeO_{4}$	34.74°	1 2 1	2.580	337.59	-	-	314.93 Å	-	359.30 Å
$Mn_{5}Ge_{3}$	35.46°	0 0 2	2.530	226.114	-	-	-	-	-
$Mn_{5}Ge_{3}$	38.24°	2 1 0	2.352	226.114	-	-	-	-	-
$Mn_{5}Ge_{3}$	38.38°	1 0 2	2.343	226.114	-	-	-	-	-
$Mn_{5}Ge_{3}$	42.35°	2 1 1	2.132	226.114	-	-	376.85 Å	370.46 Å	512.94 Å
$Mn_{5}Ge_{3}$	43.61°	3 0 0	2.074	226.114	-	-	-	-	-
$Mn_{5}Ge_{3}$	43.73°	1 1 2	2.066	226.114	-	-	-	-	-
$Mn_{5}Ge_{3}$	46.22°	2 0 2	1.963	226.114	-	-	-	-	-
${\mathrm{Mn}}_{5}({\mathrm{Si}}_{0.3}{\mathrm{Ge}}_{0.7}{)}_{3}$	35.80°	0 0 2	2.487	218.45	-	-	-	-	-
${\mathrm{Mn}}_{5}({\mathrm{Si}}_{0.3}{\mathrm{Ge}}_{0.7}{)}_{3}$	38.37°	2 1 0	2.331	218.45	-	-	-	-	-
${\mathrm{Mn}}_{5}({\mathrm{Si}}_{0.3}{\mathrm{Ge}}_{0.7}{)}_{3}$	42.53°	2 1 1	2.111	218.45	-	-	317.68 Å	-	357.84 Å
${\mathrm{Mn}}_{5}({\mathrm{Si}}_{0.3}{\mathrm{Ge}}_{0.7}{)}_{3}$	44.03°	1 1 2	2.039	218.45	-	-	-	-	-
$Mn_{1}1Ge_{8}$	39.32°	3 5 1	2.290	1055.77	-	-	-	-	-
$Mn_{1}1Ge_{8}$	41.49°	6 1 0	2.175	1055.77	-	-	-	-	-
$Mn_{1}1Ge_{8}$	42.26°	2 7 0	2.137	1055.77	-	-	-	-	-
$Mn_{1}1Ge_{8}$	42.73°	6 2 0	2.115	1055.77	-	-	-	-	333.90 Å
$Mn_{1}1Ge_{8}$	44.69°	3 3 2	2.026	1055.77	-	-	-	-	-
$Mn_{1}1Ge_{8}$	45.50°	4 1 2	1.992	1055.77	-	-	-	-	-

## Data Availability

Data will be available upon request.

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
