# Peer review of "Characterization of Mn5Ge3 Contacts on a Shallow Ge/SiGe Heterostructure"

_nanomaterials, 2024, doi:10.3390/nano14060539_

Round 1
Reviewer 1 Report
Comments and Suggestions for Authors
The manuscript reports on developing Mn5Ge3-based contacts on Ge/SiGe quantum well heterostructure via solid-state synthesis. The topic is technologically relevant. However, the post-growth metal contact deposition and processing as well as the ex-situ characterizations approach leaves questions regarding the substrate surface chemistry and its role in the contact interface formation. Which BTW, as the authors pointed out, is not well developed. It is expected that the order of Mn and Ge adatoms arriving on the Ge/SiGe surface determines the interface elemental stoichiometry, crystalline quality, and film thermal stability. These issues need to be addressed upfront before perusing/adopting the proposed approach for more specific spin injection, transport, spin lifetime investigations, and coherence etc.
We agree that the formation of Mn5Ge3 alloy using solid-state reaction is appealing, however, keeping the ultra- or even moderately thick-thin film geometry for achieving its ferromagnetic ground state with high Tc is a big challenge using the proposed approach and this report confirmed that reporting on the cross-sectional STEM & EDS analysis of Mn5Ge3-based contacts for Ge/SiGe H-FET devices leaving room for speculations.
Furthermore, why Mn layer with 100 nm thickness was selected? We believe that there might be the Mn layer thickness dependence affecting structural, electrical, and potentially also magnetic properties of Mn5Ge3 thin films with different Mn layer thicknesses. Also, it would be interesting to report/consider if the Mn layer surface roughness affects the formation of larger grains which in turn will have a profound effect on the magnetic properties of the contact.
Finally, the Mn5Ge3 Schottky barrier height of 0.10(1) eV should be compared more broadly and comprehensively with more recent reports than Ref.34.
We believe that the manuscript is premature to be published in the present form and requires more characterization and in-depth discussion.
Comments on the Quality of English LanguageMinor editing of English language required.
Reviewer 2 Report
Comments and Suggestions for Authors
This review deals Characterization of Mn5Ge3 Contacts on a Shallow Ge/SiGe Heterostructure.
Your work is very interesting for researchers. For next process, author should response to questions and improve the manuscript.
1. Introduction section, I recommend that authors would show originality compared to other researches such as Mn and Ge Heterostructure.
2. In figure 1, I recommend that authors would show more information such as the lattice parameters, unit-cell volume, crystallite size (D), and plane etc from XRD data.
3. I recommend you would increase font size in the all figures to understand data more easily.
4. From electrical properties of samples, you showed properties of Schottky diode in figure 4, is it right?
5. In figure 5, when Gate voltage was increasing, Why N was decreased? Could you explain how much energy gap of each layer in junction structure?
6. If possible, I recommend you would other data.
Comments on the Quality of English Language
I recommend Authors would check error and hypo in your manuscript.
Reviewer 3 Report
Comments and Suggestions for Authors
In this article, the authors investigate creating Mn5Ge3-based contacts on Ge/SiGe quantum well heterostructure via solid-state synthesis. XRD spectra, HRSTEM imaging and EDS map have been carried out to verify the correct Mn5Ge3-based phase formation. Schottky diode measurements, transmission line measurements, and Hall measurements reveal that Mn5Ge3-based contacts serve as good p-type contacts for Ge/SiGe quantum well heterostructures due to having a low Schottky barrier height of 0.10 eV and contact resistance on the order of 1 kΩ. The electrical characteristics have a gate-voltage dependence thereby providing tunability. I would like to give some comments and suggestions. The detailed comments are as follows:
1. What is the meaning of “Al K-red, Au L-green, Ti K-blue, O K-red, Mn K-green, Ge L-blue, Mn K, O K, Ge L and Si K” in figure 2?
2. In line 140, the authors indicate that the Region III appears to have the best film quality. How did the authors get the conclusion that the Region III has the best film quality?
3. Figure 3d shows the simulated SAED pattern of Mn5Ge3 in the [1,-2,0] projection. However, no explanation of the simulation setup information is provided, please include that.
4. Some mistakes are found, for example, in line 161, “with a resistivity of 1-10Ωcm”; in figure 4(a), “KB·T·ln(J/A·T2)”. Please proofread the manuscript carefully.
Reviewer 4 Report
Comments and Suggestions for Authors
This manuscript reported the characterization of Mn5Ge3 contacts on a shallow Ge/SiGe heterostructure. They observed the correct phase of Mn5(SixGe1−x)3 to make a p-type ferromagnetic contact for Ge/SiGe heterostructure field effect transistors (HFETs). They found evidence of Schottky barrier lowering, shown from decreasing contact resistance past saturation. However, some issues should be addressed adequately before I fully support the publication of the manuscript in nanomaterials.
1. The statement on lines 129 to 133 " The Si map shows only trace amounts where there are large concentrations of Ge. The Ge map shows that there are islands of Ge mostly near the region I/region II interface. Interestingly, the Mn and O distributions are nearly uniform in region II except at the areas of highest Ge concentrations and where the EDS data does not correspond to any of the elements tested (black region)." However, in Figure 2 i),The content of Si is higher than Ge at the interface of II and III regions. Furthermore, authors should analyze the state of the black region and whether it will affect subsequent measurements.
2. The measurement of Schottky barrier height is obtained from the Mn5Ge3/Ge Schottky diodes, not the Ge/SiGe quantum well heterostructures in the Abstract, authors should analyze the difference between the two and the feasibility of the analogy
3. The statement on lines 202 to 204 "…the contact resistance has a linear dependence on gate voltage after saturation of the device." However, in Figure 5 a),The rate of decreasing of RC increases as the gate voltage approaches -20V. Authors should explain the reasons for this change.
Round 2
Reviewer 1 Report
Comments and Suggestions for Authors
The authors have revised the manuscript satisfactorily and answered all my inquiries, thank you. I recommend publishing it as is.
Author Response
Thank you for your recommendation to publish as is.
Reviewer 2 Report
Comments and Suggestions for Authors
The author responded to all comments and questions.
I recommend you would revise font size to be large in figures.
Comments on the Quality of English Language
I recommend Authors check typo and missing in your manuscript.
Author Response
Thank you for your suggestions. We have now increased all font sizes in each of the figures to the extent possible. Additionally, we have fixed typos regarding subscripts in the captions of both Figures 2 and Figure 3.